# Numerical Simulation of Thermo-Optic Effects in an Nd: Glass Slab with Low Thermally Induced Wavefront Distortion

Xiaoqin Wang [1,2], Jiangfeng Wang [1,*], Jiangtao Guo [1], Xinghua Lu [1], Yamin Wang [1,2], Qi Xiao [1,2], Wei Fan [1] and Xuechun Li [1]

1   National Laboratory on High Power Laser and Physics, Shanghai Institute of Optics and Fine Mechanics, Chinese Academy of Science, Shanghai 201800, China; xqwang@siom.ac.cn (X.W.); guojiangtao@siom.ac.cn (J.G.); luxingh@foxmail.com (X.L.); wym@siom.ac.cn (Y.W.); xiaoqi@siom.ac.cn (Q.X.); fanweil@siom.ac.cn (W.F.); lixuechun@siom.ac.cn (X.L.)
2   Center of Materials Science and Optoelectronics Engineering, University of Chinese Academy of Science, Beijing 100049, China
*   Correspondence: wajfeng@163.com

**Abstract:** A gain slab configuration with a low thermally induced wavefront distortion, which is based on heating the edge by the cladding layer, is proposed. The gain slab will be applied to a helium-cooled Nd: glass multislab laser amplifier with an output of 100 J at a repetition rate of 10 Hz. Additionally, a 3D numerical simulation model is developed to analyze the thermo-optic effects in the gain slab. Some parameters, including the absorption coefficient ($\alpha$) of the cladding layer, the shape of the pump beam, and the gap between the pump area and absorbing cladding layer, are optimized to eliminate the thermo-optic effects. The results indicate that the peak-to-valley (P-V) of the thermally induced wavefront distortion of the specific gain slab can be reduced by 61% if other parameters remain constant.

**Keywords:** wavefront distortion; thermo-optic effects; laser amplifier; Nd: glass slabs; absorbing cladding layer





## 1. Introduction

High-power laser drivers are the core infrastructures for high-energy-density physics and fusion research. Upon constructing giant laser drivers that were presented by the National Ignition Facility (NIF) in the United States [1] and the consequent deepening of physics experiments [2], the drive facilities were expected to transition from a single-shot high-energy system to a high-repetitive-rate, high-energy drive. Moreover, next-generation petawatt (PW) lasers are expected to achieve tens to hundreds of kW average power for promising applications [3,4]. This implies that the repetition rates of the pump sources of the PW lasers must be increased [5]. High-energy, high-average-power (HE-HAP) diode-pumped solid-state laser (DPSSL) systems are good candidates in these application areas.

Efficient heat management and the reduction of thermo-optic effects dominate the design considerations of HE-HAP laser systems. Various laser head geometries, such as zig-zag slabs [6], active mirrors [7], and multislab geometries [8–10], for effectively removing waste heat from the laser medium have recently been developed. Research demonstrates that the configuration of multislabs exhibits good prospects in solid-state laser amplifiers with high energy at high repetition rates. The gain medium in a multislab laser system is divided into several discrete thin slabs. There are gaps between the adjacent slabs, and a high-speed coolant flows through such gaps to facilitate efficient heat exchange between the slabs and coolant, thus achieving an increased repetition rate. Further, the Lawrence Livermore National Laboratory (LLNL), USA, built a Mercury laser facility for the development of inertial fusion energy (IFE) [11,12], and the facility was operated continuously for several hours at 55 J and 10 Hz in 2008 [13,14]. A 100 J-class multislab

laser system, the DiPOLE laser, developed and built at the STFC Rutherford Appleton Laboratory and now installed at HiLASE Centre (Czech), was operated at 105 J, 10 ns pulses, and 10 Hz in 2017 [15,16]. The gain media in the project were surrounded by the absorptive cladding to minimize the influence of amplified spontaneous emission (ASE) on laser performance [17]. Furthermore, the HAPLS project notably achieved an output of 97 J at 3.3 Hz in 2017, employing Nd: glass as the gain medium in the laser system [18,19]. The thermo-optic effects, including wavefront distortion [20], thermal depolarization [21], and thermal lensing effects [22], are inevitable in the operation of the laser amplifier. For a multislab gas-cooled laser amplifier, the temperature gradient is mainly concentrated around the pump area, and the strong temperature field leads to wavefront distortion, which is difficult to compensate. There are some reports about theoretical calculations of thermo-optic effects in the Yb: YAG gain medium with a cladding layer [23,24], but the sizes of pump area in the literature are the same as the Yb: YAG gain medium, which are in ideal cases and require the extremely high performance of the beam alignment. Beam misalignment would cause high-order wavefront distortion, which is difficult to compensate [25]. Actually, the size of the pump area is usually smaller than the gain medium when considering the coupling efficiency and experimental operation difficulty. Some numerical analyses and experimental studies for already-existing configurations have been developed [26], and the preliminary optimization of cladding geometry for reducing thermal stress-induced depolarization in high-energy, high-repetition-rate, diode-pumped Yb: YAG lasers is also included in the literature [27]. The results show that the design for the cladding is effective to reduce thermal depolarization. Above all, it is necessary to study the thermo-optic effects, especially wavefront distortion in specific cases.

The gain medium is also vital in a laser amplifier system. The materials for laser operations, e.g., Yb: YAG [7], Nd: YAG [28], and Nd: glass [29], have been extensively investigated. For high-energy laser applications, an Nd: glass with good spectroscopic properties, high solubility (characteristic of rare-earth ions), and ease of being produced in large sizes is required. The LG-770 and LHG-8 laser glasses are utilized in the NIF and LMJ lasers for inertial confinement fusion research. In China, the N31 glass (Nd: glass) has been developed at the Shanghai Institution of Optics and Fine Mechanics and utilized in the SG-II [30] and SG-III laser facilities [31]. Furthermore, laser amplifiers employing Nd: glass possess increased capability toward short-duration pulse Texas Petawatt (TPW) Lasers [32].

In this paper, a special gain medium architecture with low thermally induced wavefront distortion is proposed based on the multislab geometry. The gain slab consists of a solid-state laser material and a cladding layer. The amplified spontaneous emission (ASE) energy absorbed in the cladding layer is converted into heat and employed as the heat source for edge heating, therefore, reducing the thermo-optic effects. The gain slab would be applied to a multislab helium-cooled 100 J/10 Hz Nd: glass laser amplifier system. We evaluate the thermo-optic effects of the gain slab via a 3D numerical simulation model. Some parameters, including the absorption coefficient of the cladding layer, the shape of the pump beam, and the gap between the pump area and absorbed cladding layer, are optimized in the numerical model to reduce the thermo-optic effects of the gain slab. The simulation results indicate that the peak-to-valley (P-V) of the thermally induced wavefront distortion of the specific gain slab can be reduced by 61% if other parameters remain constant.

## 2. Theoretical Simulation

The waste heat in the gain slabs is caused mostly by the Stokes defect, concentration quenching, and upconversion [33]. Only for a strong pump intensity or heavy doping concentration do concentration quenching and upconversion need to be considered for the contribution to thermo-optic effects [34]. The combination of the heat production of the gain medium that is associated with the absorbed pump radiation and surface cooling causes the temperature gradient around the pump area. The temperature gradient

can consequently cause thermal stress and deformation. Moreover, it can change the refractive index coefficient characterized by the thermo-optic coefficient. The refractive index can also be changed by thermal stress and the different polarizability of the excited ions in the excited state and ground state, and the latter will not be ignored only for a high doping concentration [35,36]. When a signal laser passes through the gain medium, accompanied by heat deposition, it induces an optical path difference (OPD), which would severely degrade the optical quality of the laser beam and eventually limit the laser output power. We developed a 3D model for exploring the thermo-optic effect of the gain slab, including heat deposition distribution, temperature, stress and deformation distribution, and wavefront distortion.

The heat generated in the Nd: glass slab is mainly due to the quantum defect, and the heat deposition in the cladding is converted by absorbing the energy of the ASE in the Nd: glass. The heat distribution in the Nd: glass is calculated according to the pump power distribution absorbed. The fraction of power converted into heat in the pumped volume as a result of the quantum defect is: $P_{QD} = P_{Pump}(1 - \frac{\lambda_P}{\lambda_L})$, where $P_{QD}$ is the power of waste heat, $P_{Pump}$ is pump power, and $\lambda_P$ and $\lambda_L$ are the wavelength of the pump and laser beam, respectively. The heat deposition in cladding is related to the ASE effects of the amplifier, and detailed descriptions of the ASE effects will be published in another paper. The heat distribution in the cladding would be changed for different absorption coefficients of cladding and pump parameters.

For an ns-level pulse-pumped solid-state laser amplifier, the laser pulse interval is much less compared to the thermal relaxation time. Thus, the laser amplifier is generally operated in a steady state. The temperature distribution, $T(x, y, z)$, could be solved by the steady-state heat conduction equation:

$$\nabla^2 T(x, y, z) = -\frac{Q(x, y, z)}{\kappa},$$

(1)

where $Q(x, y, z)$ is the thermal power density (W/cm$^3$), and $\kappa$ is the thermal conductivity. The 3D temperature distributions can be solved according to heat deposition in the gain slab and some boundary conditions of the laser head. The 3D structure of the two gain slabs with mounting vanes and the cooling channel is shown in Figure 1. The four edges of a gain slab are assumed to be thermal adiabatic, which means that there are no heat exchanges between the edges of the gain slab and mounting vane. This assumption is reasonable because each edge has a small area and litter heat transfer with mounting vane. The two surfaces of a gain slab exchange heat with high-speed helium-flow in the cooling channel. The heat transfer coefficient is determined by the properties of the turbulent flow in cooling helium.

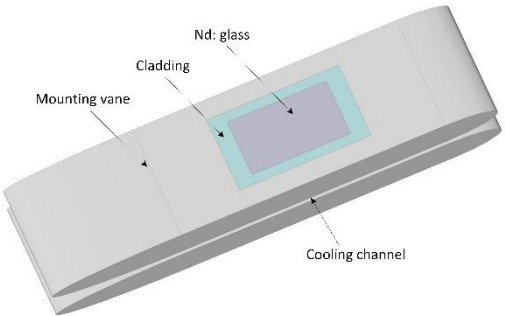

**Figure 1.** Three-dimensional (3D) structure diagram of the two gain slabs with mounting vanes and a cooling channel.

In a nonuniform temperature field, the medium would change the volume and generate stress therein. The thermal stress and deformation in a gain slab can be obtained by solving equations of elasticity, which indicate relationships between temperature, stress,

and deformation. The changes in the refractive index can also be calculated with the thermo-optic coefficient and photoelastic effects. Regarding the deformation, $U(x, y, z)$, it could be expressed under the force balance condition, as follows:

$$\nabla^2 U(x,y,z) + \frac{1}{1-2\nu}\nabla(\nabla \bullet U(x,y,z)) = \frac{2(1+\nu)}{1-2\nu}\alpha\nabla T(x,y,z), \quad (2)$$

where $\nu$ is Poisson's ratio, and $\alpha$ is the thermal expansion coefficient.

The components of the strain tensor, $\varepsilon_{ij}(x, y, z)$, can be expressed regarding the derivatives of the displacement [37], as follows:

$$\varepsilon_{ij}(x,y,z) = \frac{1}{2}\left[\frac{\partial U_j(x,y,z)}{\partial x_i} + \frac{\partial U_i(x,y,z)}{\partial x_j}\right] \quad (i, j = 1, 2, 3). \quad (3)$$

According to the generalized Hooke's law, the stress tensor components $\sigma_{ij}(x, y, z)$ are linearly dependent on the strain tensor components in the linear-elastic materials, which can be written in the following form:

$$\sigma_{ij}(x,y,z) = \frac{E}{1-\nu}\left[\varepsilon_{ij}(x,y,z) + \frac{\nu}{1-2\nu}(\varepsilon_{xx}(x,y,z) + \varepsilon_{yy}(x,y,z) + \varepsilon_{zz}(x,y,z))\delta_{ij}(x,y,z) - \frac{1+\nu}{1-2\nu}\alpha T(x,y,z)\delta_{ij}(x,y,z)\right], \quad (4)$$

where $E$ is the Young's modulus (Pa), and $\delta_{ij}$ is the Kronecker delta.

When a temperature gradient, thermal stress, and deformation exist in the gain medium, the laser optical path that the laser beam passes through in the gain medium changes [38], as follow:

$$OPD(x,y) = \int_0^L \frac{\partial n}{\partial T(x,y,z)}\Delta T(x,y,z)dz + \int_0^L \sum_{i,j=1}^3 \frac{\partial n}{\partial \varepsilon_{i,j}(x,y,z)}\varepsilon_{i,j}(x,y,z)dz + (n_0 - 1)\Delta L, \quad (5)$$

where the first term on the right indicates the variation in the refractive index that depends on the temperature gradient, thereby changing the optical path, $\frac{\partial n}{\partial T(x,y,z)}$ is the thermo-optic coefficient. The second term represents the influence of the change in the refractive index on the optical path, which is caused by the thermal stress in the gain medium. The third term indicates the effect of the deformation along the optical path, where $\Delta L$ is the thickness of the gain medium.

In fact, the calculation of temperature, thermal stress, and deformation in the gain slab is huge and complex, which can be carried out by a commercial finite-element analysis software named COMSOL.

## 3. Geometry of the Laser Amplifier

The laser amplifier is designed to generate a 100 J/10 Hz output at 1053 nm, and it can deliver an average power of 1 kW. The laser system is a DPSSL system that is based on the helium-cooled multislab configuration employing two identical power amplifier heads. The structure of a laser head, including ten square Nd: glass slabs, is shown in Figure 2a. The gain media are pumped from both ends by a laser diode (LD) pumping source with a central pump wavelength of 802 nm. In order to get a 100 J/10 Hz output, each LD pumped source delivers a pump fluence of 5 J/cm$^2$, respectively. The pump area is 56 mm × 56 mm (full-width at half-maximum (FWHM)). The average pump power for a LD pumping source is about 1500 W, which is the product of the total pump energy of a LD source and the repetition rate. Therefore, the total average pump power for a laser head is 2700 W, which is calculated based on the estimate of the transmission efficiency of the pumped coupling system with 90%. The cross-section of a gain slab normal to laser propagation is displayed in Figure 2b, and the pump area is in the dashed line. The outer dimensions of each slab are 80 mm × 80 mm × 10 mm. The Nd: glass of 60 mm × 60 mm is surrounded by a 10-mm-thick absorbing cladding layer, which is used to convert ASE energy to heat for edge heating. The ten Nd: glass slabs have uniform but different doping

concentrations, which are designed to ensure that the pump energy absorbed by each slab is similar, thereby making the distribution of heat density for the ten slabs roughly uniform. The average heat-power density in each Nd: glass slab is about 2 W/cm³ due to the Stokes defect, but it varies in the cladding with the absorption coefficient of cladding. The helium temperature at the inlet of the laser head is 293 K, and the cooling channels between the adjacent Nd: glass slabs in the laser amplifier are 1 mm. The helium pressure is fixed at 4 atm and its volume flow rate is 3 m³/min. The thermal dissipation of the gain slab is estimated by a uniform convective heat transfer coefficient of 1600 W/m²/K.

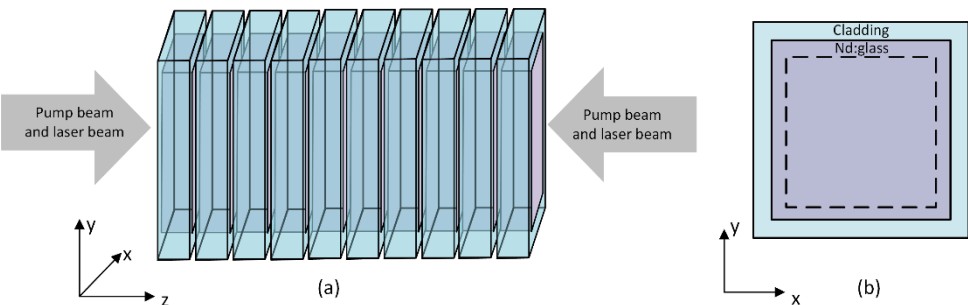

**Figure 2.** (**a**) Configuration of a multislab Nd: glass laser head and (**b**) a diagram of a monolithic gain slab. The pump area is in the dashed box.

The absorption coefficient ($\alpha$) of the absorbing cladding layer, the gap between the pump area and the absorbing cladding layer, and the shape of the pump beam must be optimized according to the thermo-optic effects. The thermal and optical parameters of Nd: glass [31] and the absorbing cladding layer [39] are presented in Table 1.

**Table 1.** Main parameters of Nd: glass and the cladding layer.

| Properties | Nd: Glass | Cladding Layer |
|---|---|---|
| Stimulation emission cross-section/($10^{-20}$cm²) | 3.8 | - |
| Fluorescence lifetime/µs | 351 | - |
| Density/(g/cm³) | 2.87 | 2.86 |
| Refractive index at the lasing wavelength | 1.533 | 1.5306 |
| Thermo-optical coefficient (dn/dT)/($10^{-7}$K$^{-1}$) | −43 | - |
| Coefficient of the thermal expansion/($10^{-7}$K$^{-1}$) | 116 | 112.7 |
| Thermal conductivity/(W·m$^{-1}$·K$^{-1}$) | 0.56 | 0.56 |
| Specific heat/(J·g$^{-1}$·K$^{-1}$) | 0.75 | 0.75 |
| Young's modulus/(GPa) | 56.4 | 52.7 |
| Poisson's ratio | 0.26 | 0.27 |

## 4. Principle of Reduction of the Thermo-Optic Effects

In a traditional multislab laser amplifier, the pump area is much smaller than the gain medium, and the cladding layer is designed for the purpose of the protection of the slab from the oscillation of the ASE [40]. In this paper, the size of the pump area is close to the gain medium, and the heat deposition in the cladding layer is the heat source for edge heating. The temperature gradient around the pump area can be decreased by edge heating, therefore reducing the wavefront distortion in the gain medium. In order to illustrate the principle of reduction of the thermo-optic effects for the special gain slab, we took a single gain slab as an example to investigate its heat density, temperature, stress, and wavefront distribution and compared these thermo-optic properties with that of an Nd: glass slab without the absorbing cladding layer. The configuration of a single gain slab with the absorbing cladding layer is shown in Figure 3a. The gain slab without the absorbing cladding layer is presented in Figure 3b. In order to ensure the consistency of the structure of the gain slab, the dimensions of the two gain slabs were 80 mm × 80 mm

× 10 mm. The width of the absorbing cladding layer was 10 mm. The pump areas were both located in the center of the gain slabs, and their sizes were close to 60 mm × 60 mm. Considering the overlap efficiency between the pump and laser beams, a laser beam of 56 mm × 56 mm was chosen.

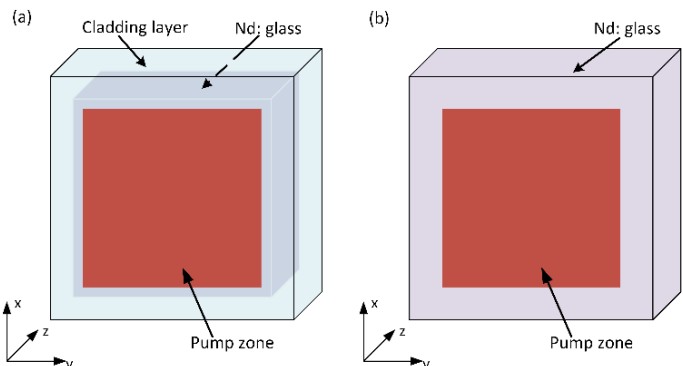

**Figure 3.** Structure of the gain slab (**a**) with the absorbing cladding layer and (**b**) without the absorbing cladding layer.

### 4.1. Heat Density Distribution

To evaluate the thermal effect of the special gain slab geometry, the heat load distributions in the Nd: glass slab and absorbing cladding layer were first calculated according to the pump absorption and ASE effect. The heat density distributions of the pump area and the absorbing cladding layer were relevant to $\alpha$ of the absorbing cladding layer, pump intensity distribution, and the gap between the pump area and the absorbing cladding layer. It was assumed that $\alpha$ of the absorbing cladding layer was 4/cm. The pump area is 56 mm × 56 mm (full-width at half-maximum (FWHM)), The deposited heat densities of the Nd: glass slab and absorbing cladding layer are shown in Figure 4. The dotted lines indicate the laser beam area in Figure 4a,b.

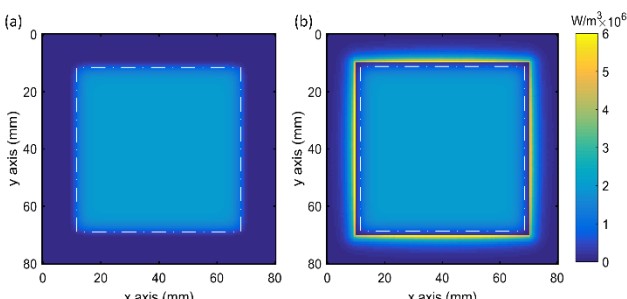

**Figure 4.** Heat density distribution in the gain slab (**a**) without and (**b**) with the absorbing cladding layer, and the laser beam area is indicated in the dashed frame on the drawing.

As shown in the figure, without the cladding layer, the heat was just deposited in the pump area (Figure 4a). However, regarding the gain slab with cladding, the heat was also deposited in the absorbing cladding layer beside the pump area (Figure 4b). This enabled volumetric heating from the cladding layer. The heat density distribution in the absorbing cladding layer decreased exponentially from inside to outside of the cross-section of the absorbing cladding layer. The heat density at the interface between Nd: glass and the absorbing cladding layer was 6 W/cm$^3$, which was up to three times higher than that at the central area of the pump beam of 2 W/cm$^3$. Heat deposited in the absorbing cladding layer was served as the heat source of edge heating.

### 4.2. Temperature and Stress

Based on the heat density distributions, finite-element analysis was applied to calculate the temperature and stress. The associating temperature distributions at the central cross-sections of the two slabs are displayed in Figure 5.

Figure 5 shows the temperature distribution in the central cross-section (z = 5 mm) and the surface of the slabs without and with the absorbing cladding layer. Figure 5a,b shows the temperature distribution in the central cross-section of the two slabs, and Figure 5c,d shows those at the surface (z = 0 or z = 10 mm) of the two designs. The dotted lines indicate the laser beam areas for the four figures. The temperature distributions were similar to the heat density depositions for the two cases. However, the temperature gradient around the laser beam area of the slab with the absorbing cladding layer was less than that in the slab without the absorbing cladding layer owing to the edge heating by the absorbing cladding layer. Regarding the central cross-sections of the two designs, the temperature difference was 40 K for the gain slab without the absorbing cladding layer and 22 K for the slab with the absorbing cladding layer, as shown in Figure 5a,b. Furthermore, it can be seen from Figure 5c,d that the maximum temperature at the surfaces for the two structures (300 K) are lower than those at the central cross-sections (347 K), which is because that high-speed helium flows through the cooling channel to remove the waste heat at the surface of the gain slab, whereas the central cross-section of the gain slab is just cooled by heat conduction. Similarly, the temperature differences at the slab surface (3.6 K for slab without the cladding and 1.9 K for slab with the cladding) are much smaller than those in the central cross-section. Therefore, the edge heating by the absorbing cladding layer remarkably decreased the temperature difference in the laser beam area, and helium cooling is an efficient thermal management technology for reducing the thermo-optic effects.

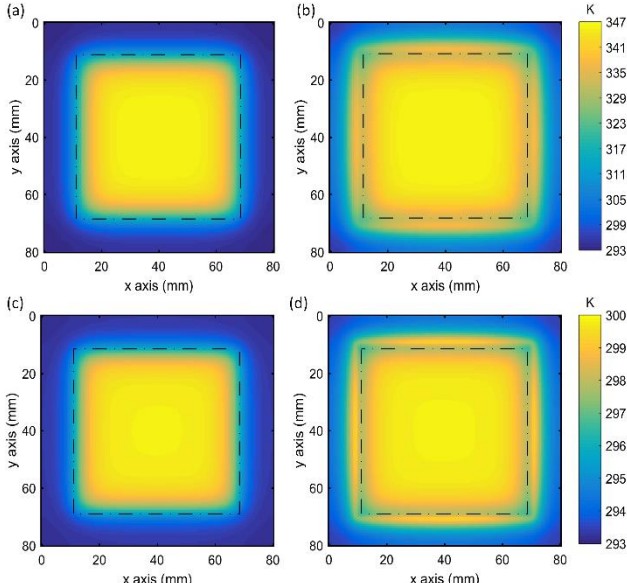

**Figure 5.** Temperature distributions in the central cross-section of the gain slab (**a**) without and (**b**) with the cladding glass, and those at the surface of the gain slab (**c**) without and (**d**) with the cladding glass. The dotted lines indicate the laser beam areas.

Figure 6 illustrates the shear stresses of the two gain slabs, and the dotted lines indicate the laser beam area. As illustrated in the figures that the maximums of the shear stresses were located at the four corners of the pump areas of both slabs. It was clear that the distribution of the shear stress was more uniform in the slab with the absorbing cladding layer because of the smaller temperature gradient therein. The maximum shear stresses of the laser beam area were 5.94 MPa for the slab without the absorbing cladding layer and 2.22 MPa for the slab with it.

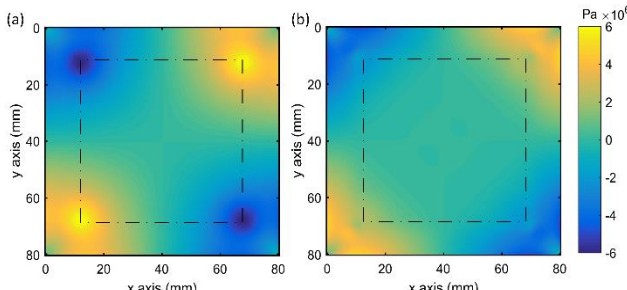

**Figure 6.** Mean stress distributions in the central cross-sections of the gain slabs (**a**) without and (**b**) with the absorbing cladding layer. The dotted lines indicate the laser beam area.

### 4.3. Wavefront Distribution

The OPD distribution for a single pass through a single gain slab is shown in Figure 7. OPD was associated with the temperature, stress, and deformation in the gain slab. The figure shows only the OPD of the laser beam area.

The distribution of the wavefront in the center area of the laser beam area was uniform. Nevertheless, the rapid change of the wavefront distortion was distributed at the edge of the laser beam area, and this was because of the temperature gradient on the fringe of the laser beam area due to the transverse heat dissipation from the gain slab. Notably, the change of the OPD was faster in the slab without the absorbing cladding layer than that with the cladding layer. The calculated maximum of the OPD of the laser beam area was $1.51\,\lambda$ for the slab without the absorbing cladding layer and $0.92\,\lambda$ for the slab with the composite structure. Thus, the gain slab architecture could effectively diminish the thermo-optic effects.

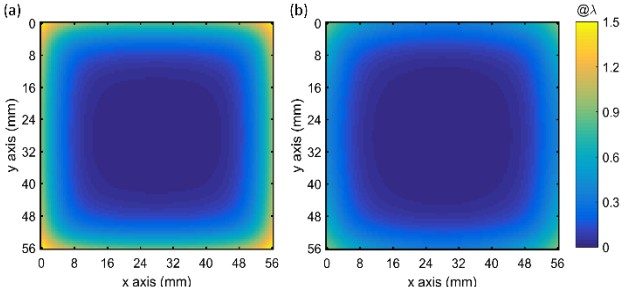

**Figure 7.** The calculated optical path difference (OPD) in the laser beam area of the gain slab (**a**) without and (**b**) with the absorbing cladding layer.

## 5. Optimal Design of the Gain Slab Architecture

### 5.1. $\alpha$ of the Absorbing Cladding Layer

The heat-power densities are different for the absorbing cladding layer with different $\alpha$. Thus, the temperature gradient changes with an $\alpha$ of the absorbing cladding layer.

The numerical model simulated the temperatures, stresses, and OPDs of several Nd: glass slabs with different $\alpha$. The model presumed that the pump beam was 56 mm FWHM wide with a square-shaped, flat-top profile and located at the center of the gain slab, and the softening factor was 0.1. $\alpha$ was changed from 0 to 6/cm. $\alpha = 0$ indicated that the cladding layer around Nd: glass did not absorb ASE (i.e., without the absorbing cladding layer, as shown in Figure 3b).

The calculated temperature distributions in the cross-section of the middle of the gain slab along the centerline and the diagonal are displayed in Figure 8a,b. The dotted lines indicate the laser beam area. It was evident that the temperature of the center region of the slab was uniform, although it rapidly dropped near the edge of the laser beam area. The temperature gradient in the laser beam area decreased with $\alpha$ because the cladding absorbed ASE and converted the heat directly for edge heating. Thus, the larger $\alpha$ was, the

lower the temperature gradient would be. The temperature difference in the four corners of Nd: glass was larger compared with those in the other edge areas.

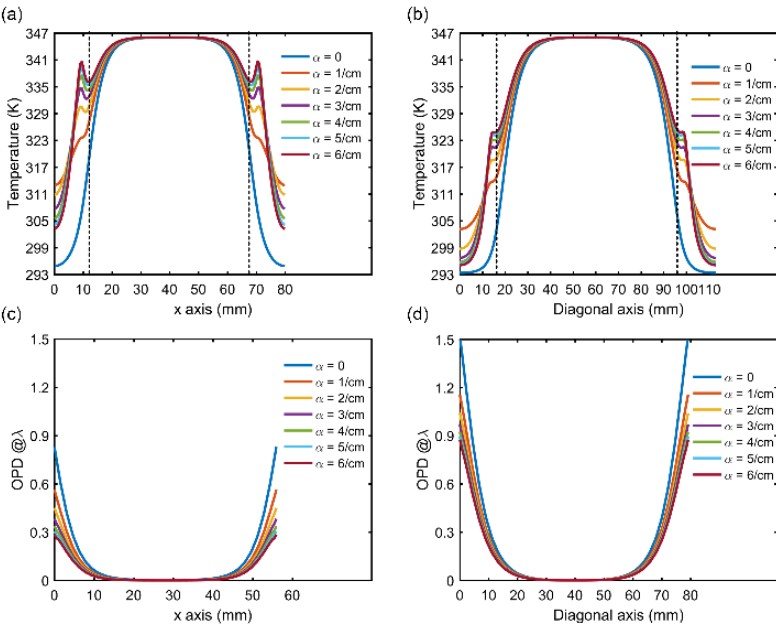

**Figure 8.** One-dimensional (1D) temperature and OPD as functions of $\alpha$, (**a**) temperature along the centerline of the gain slab with different $\alpha$, (**b**) temperature along the diagonal of the gain slab with different $\alpha$, (**c**) OPD along the centerline for a single gain slab, and (**d**) OPD along the diagonal for a single gain slab.

OPDs calculated for a single pass through a monolithic gain slab with heat deposition are shown in Figure 8c,d, respectively. Figure 8c shows a 1D OPD distribution along the centerline of the laser beam area. Figure 8d shows a 1D OPD distribution along the diagonal of the laser beam area. The wavefront distribution in the center zone of the laser beam was homogeneous, although a steep aberration emerged at the margin of the laser beam. The peak-to-valley (P-V) of the OPD reduced with $\alpha$. At $\alpha$ = 6/cm, the P-V of the OPD was 0.87 $\lambda$, 57.6% of the P-V of the OPD of the laser beam area for the slab without the absorbing cladding layer. Therefore, it was beneficial to bond the absorbing cladding layer around Nd: glass with an $\alpha$ of 6/cm to improve its nearfield output.

### 5.2. Shape of the Pump Intensity

Research has suggested that the uniformity of the pump beam significantly impacts the output energy of a laser system. A highly uniform pump distribution produces a good nearfield pattern of the output [41]. The influence of the shape of the pump beam on OPD for a gain slab was evaluated by a numerical calculation.

The 1D distribution of the pump beam is shown in Figure 9. The shape of the pump beam was characterized by the softening factor (Q). Q was defined as the ratio of the slope width, $L_1$, to the FWHM of the pump beam:

$$Q = \frac{L_1}{L},\tag{6}$$

where $L_1$ of the pump beam was defined as the width from 1% to 90% of the pump intensity, and $L$ of the pump beam was the width of 50% of the maximum of the pump intensity. FWHM rather than the width of 1% of the pump intensity was selected with the aim of obtaining a more accurate overlap efficiency between laser beam and pump beam.

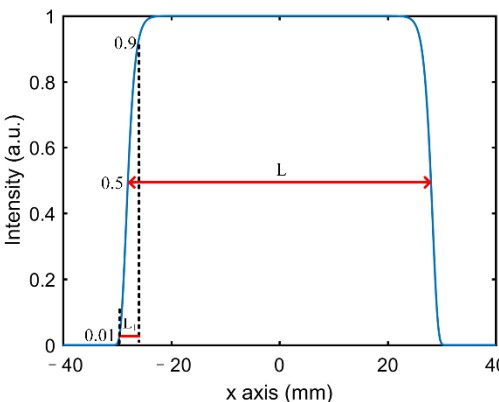

**Figure 9.** One-dimensional (1D) pump distribution of the gain slab.

Q was optimized to reduce the temperature gradient at the edge of the laser beam area. Therefore, the P-V of the OPD of the laser beam passing through the gain slab with thermal deposition could be minimized, and the results are presented in Figure 10. *L* in the model was 56 mm × 56 mm, and Q was changed from 0.02 to 0.1. A smaller Q signifies better beam quality from the pump coupling system. P-Vs of the OPD were calculated with an $\alpha$ of 0 and 6/cm.

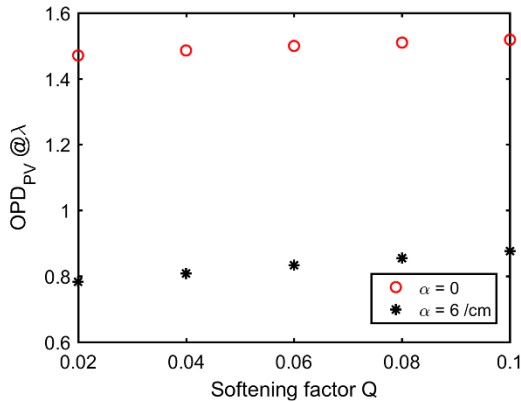

**Figure 10.** Peak-to-valley (P-V) of the OPD for a single slab as a function of Q with two absorption coefficients: $\alpha = 0$ (red) and $\alpha = 6/\text{cm}$ (black).

As shown in the figure, the wavefront distortions were much less influenced by Q for a gain slab without the absorbing cladding layer and remained around 1.5 $\lambda$. The P-V of the OPD was reduced dramatically for a gain slab with the absorbing cladding layer with an $\alpha$ of 6/cm. The P-V of the OPD was 0.88 $\lambda$ when Q was 0.1. However, it was reduced to 0.78 $\lambda$ when Q was 0.02, 51.7% of the P-V of the OPD of the gain slab without the absorbing cladding layer. Therefore, it was necessary to design a high-quality pump coupling system with a small Q of the pump beam to improve the output beam quality of a laser amplifier.

### 5.3. Gap between the Pump Area and Absorbing Cladding Layer

The gap between the pump area and the absorbing cladding layer must be optimized to achieve efficient heat removal and abate the thermal effect. The gap is defined as half of the difference between the FWHM of the pump beam and the width of the Nd: glass.

As the pump beam became smaller than the Nd: glass, it generated a gap between the pump area and the absorbing cladding layer, which would enlarge the temperature gradient because of the limited edge heating capacity. The thermo-optic effects across the laser beam area of the laser medium reduce the beam quality and the performance of the laser system. The model analyzed the influence of the gap on OPD for a gain slab. Two designs were simulated. One was a gain slab without the absorbing cladding layer.

The other was an Nd: glass bonded to the cladding layer, and the $\alpha$ of the cladding layer was 6/cm. Both the two gain slabs were pumped by LD pumping sources. Qs of the pump beams of the two gain slabs were 0.02 each. The pump areas varied according to the designed gaps.

The calculated P-Vs of the OPD within the laser beam area as functions of the gaps in the two structures are shown in Figure 11. As shown in the figure, the P-Vs of the OPD increased with the gaps. The P-Vs of the OPD were 1.90 and 1.38 $\lambda$ for the two designs of gaps of 5 mm, and they decreased to 1 and 0.38 $\lambda$ when FWHM of the pump beam equaled to the width of the Nd: glass. The P-V of the OPD was larger for the gain slabs without absorbing cladding layers compared with that they did. In fact, it was almost impossible to maintain a very small width for the gap because it would not allow any alignment tolerance. This was beyond the practical operation limits in the process of building a laser amplifier. In addition, the overlap efficiency between the pump beam and the laser beam was also needed to be considered. The laser beam area was 56 mm × 56 mm in the numerical model. The pump beam was slightly larger than the laser beam to ensure that the energy was extracted as effectively as possible. Overlap efficiencies were 98%, 91.5%, and 85.5% for the gap of 2, 1, and 0 mm, separately. The following trade-off should be considered when designing a laser amplifier: an appropriate gap between the pump area and the absorbing cladding layer versus overlap efficiency. Therefore, the gap of 1 mm was preferable, and the P-V of the OPD calculated for a single pass was 0.47 $\lambda$ (39% of the P-V of the OPD of the laser beam area for the slab without the absorbing cladding layer).

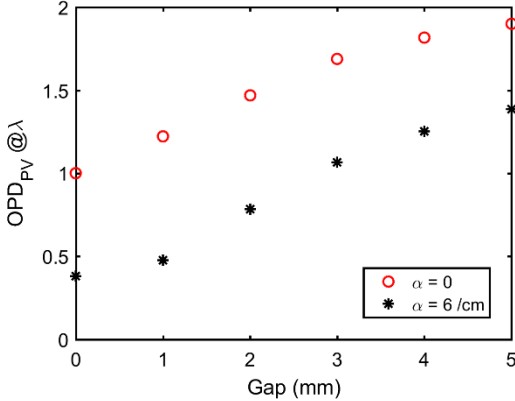

**Figure 11.** The P-V of the OPD for a single slab as a function of the gap between the pump area and absorbing cladding layer within the laser beam area of 56 mm × 56 mm in the two designs. Nd: glass without the absorbing cladding layer (red), and Nd: glass with cladding and $\alpha = 6$/cm (black).

## 6. Conclusions

A helium-cooled multislab Nd: glass laser amplifier with an output of 100 J at 10 Hz was proposed conceptually. A special gain medium configuration that could achieve low thermally induced wavefront distortion was carefully designed. The gain slab was based on edge heating by the absorbing cladding layer to reduce the temperature gradient, thereby reducing the wavefront distortion. The results of the numerical analysis demonstrated that OPD was lower in the slab with the absorbing cladding layer compared with that without it when the laser beam passed through the Nd: glass slab. The gain slab would be used in the100 J/10 Hz Nd: glass laser amplifier.

For an Nd: glass slab (10 mm thick) that was surrounded by an absorbing cladding layer with an $\alpha$ of 6/cm, the heat-power density was 2 W/cm. The size of the pump beam was 58 mm × 58 mm (FWHM), and Q was 0.02. The calculated P-V of the OPD in the laser beam area of 56 mm × 56 mm was 0.47 $\lambda$, which was much smaller compared to that (1.22 $\lambda$) in the slab without the absorbing cladding layer. The P-V of the thermally induced wavefront distortion was reduced by 61%. Evidently, the gain slab geometry favored the reduction of the thermo-optic effects. Therefore, the gain slab, which achieved a low

thermally induced wavefront distortion, is promising and can be applied to a multislab amplifier with a high-energy and high repetition rate.

According to the numerical analysis, for the 100 J/10 Hz Nd: glass laser amplifier comprised two power amplifier heads; if twenty square-shaped Nd: glass slabs were all not surrounded by the absorbing cladding layers, the P-V of the OPD for the laser amplifier would be 24.4 $\lambda$. It would be decreased to 9.4 $\lambda$ if the special gain slabs were employed in the laser amplifier. Residual thermal aberrations within the amplifier head can be compensated further by the adaptive optics system to improve output beam quality.

Based on the results of the numerical simulation, a prototype helium-cooled multislab Nd: glass laser amplifier is developed, which will be capable of amplifying nanosecond pulses to an energy of 100 J at 10 Hz. The success of the work will demonstrate the feasibility and validity of the conceptual design and provide designing ideas and references for the next kJ-class laser system.

**Author Contributions:** Conceptualization, X.W., J.W. and X.L. (Xuechun Li); methodology, X.W., J.W., J.G., and X.L. (Xuechun Li); investigation, X.W., J.W. and J.G.; Writing original draft preparation, X.W.; writing review and editing, X.W., J.W., J.G., X.L. (Xinghua Lu), Y.W. Q.X., W.F. and X.L. (Xuechun Li); funding acquisition, J.W., W.F. and X.L. (Xuechun Li); project administration, J.W. and W.F. All authors have read and agreed to the published version of the manuscript.

**Funding:** This research was funded by the TECHNOLOGY RESEARCH LEADER, grant number 19XD1404000.

**Data Availability Statement:** The data presented in this study are available on request from the corresponding author. The data are not publicly available due to company proprietary information.

**Acknowledgments:** This work was supported by National Laboratory on High Power Laser and Physics in Shanghai Institute of Optics and Fine Mechanics.

**Conflicts of Interest:** The authors declare no conflict of interest.

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
