# Peer review of "Numerical Simulation of Thermo-Optic Effects in an Nd: Glass Slab with Low Thermally Induced Wavefront Distortion"

_photonics, doi:10.3390/photonics8040091_

Round 1

Reviewer 1 Report

See attached pdf file.

Author Response

Dear reviewer

Thank you for your comments, we have studied comments carefully and made correction which we hope meet with approval. Please see the attachment for details. 

Reviewer 2 Report

The article is overall well written and provides interesting and relevant information. However, detailed information about the simulation techniques is missing. The authors just display the equations but do not explain which tools are employed to solve them and how they are employed.

Line 59: order

Line 74: The last paragraph of section 1 is confusing. Due to the usage of simple past tense, it is not totally clear (at
least not at the beginning of the paragraph) if the authors refer to their paper or the reference [28]. Generally there
are some weird phrases in this section such as "The numerical model optimized some parameters" - it did not, it may have been
used for optimization, though. I recommend proofreading this section again.

Section 2:

I miss a figure illustrating the system that is being simulated here. Whithout that, the reader can hardly understand
the explanations.

What techniques do the authors employ for the simulation? Commercial tools like COMSOL? Is it based on FEA or other 
techniques? There is a lack of technical detail in this section. At least, the heading "theoretical simulation" implies more than 
writing down the fundamental governing equations. It does not become clear how these equations are solved. The first time I found 
any hind on that was on page 5.

For the overall understanding it may be helpful to flip the order of sections 2 and 3.

The authors refer to an "absorbed cladding layer" which is used in order to mitigate the wave front distortion in their system.
Shouldn't it be "absorbing cladding layer"?

Author Response

(The authors gave the same response as above.)

Reviewer 3 Report

Xiaoqin Wang and co-authors proposed and numerically studied a gain slab configuration with low thermally induced wavefront distortion for application to helium-cooled Nd: glass multi-slab laser amplifier with an output of 100 J at a repetition rate of 10 Hz. The authors focused of mitigation of thermo-optical effects, but they ignore the effect related to refractive index changes caused by population changes of ground and excited states of Nd3+ ions having different polarizabilities. However, this effect can give a significant contribution to wavefront distortion (see Ref. [*]). As said in the abstract in Ref. [*]: “The results indicate that, for the case of a single-shot laser amplifier system, the beam distortion induced by this population-induced change in the refractive index is comparable with the distortion arising from thermal effects.”

The detailed numerical model for describing refractive index changes in isotropic matrix doped with rare-earth ions is presented in [**].

Further, it is well known that up-conversion processes in Nd-doped media matrices can give a noticeable contribution to the population of energy levels. In the paper by Xiaoqin Wang et al., this effect is also ignored and not mentioned, but should be described (estimate) and discussed (for example as in [***]).

Recommendations:

I. I strongly recommend to the authors to estimate contribution to their results (recalculate results if necessary) of

1) refractive index changes caused by population changes of ground and excited states of Nd3+ ions having different polarizabilities;

2) up-conversion processes in Nd-doped glass.

Refs. [*, **, ***] should be cited in the proper context.

II. In relevant Ref. [****], depolarization dynamics in a neodymium glass rod laser amplifier was experimentally studied. This paper should be mentioned in the introduction.

III. In line 128: “The Nd: glass slabs are cooled to 293K”. But 293K corresponds to room temperature, not the temperature obtained this helium-cooled system. This phrase should be revised.

[*] Powell, R. C., Payne, S. A., Chase, L. L., Wilke, G. D. Index-of-refraction change in optically pumped solid-state laser materials. Opt. Lett. 1989, 14(21), 1204-1206.

[**] Antipov, O. L., Anashkina, E. A., Fedorova, K. A. Electronic and thermal lensing in diode end-pumped Yb: YAG laser rods and discs. Quantum Electronics 2009, 39(12), 1131.

[****] Kuz'min, A. A., Luchinin, A. G. E., Poteomkin, A. K., Soloviev, A. A., Khazanov, E. A., & Shaikin, A. A. Thermally induced distortions in neodymium glass rod amplifiers. Quantum Electronics 2009, 39(10), 895.

Author Response

(The authors gave the same response as above.)

Round 2

Reviewer 3 Report

The revised paper may be published.